# Multimodal Automated Fact-Checking: A Survey

**Mubashara Akhtar[1,*], Michael Schlichtkrull[2], Zhijiang Guo[2], Oana Cocarascu[1],**
**Elena Simperl[1] and Andreas Vlachos[2]**

[1]Department of Informatics, King's College London
[2]Department of Computer Science and Technology, University of Cambridge

{mubashara.akhtar,oana.cocarascu,elena.simperl}@kcl.ac.uk
{mss84,zg283,av308}@cam.ac.uk

## Abstract

Misinformation is often conveyed in multiple modalities, e.g. a miscaptioned image. Multimodal misinformation is perceived as more credible by humans, and spreads faster than its text-only counterparts. While an increasing body of research investigates automated fact-checking (AFC), previous surveys mostly focus on text. In this survey, we conceptualise a framework for AFC including subtasks unique to multimodal misinformation. Furthermore, we discuss related terms used in different communities and map them to our framework. We focus on four modalities prevalent in real-world fact-checking: text, image, audio, and video. We survey benchmarks and models, and discuss limitations and promising directions for future research.

## 1 Introduction

Motivated by the challenges presented by misinformation in the modern media ecosystem, previous research has commonly modelled automated fact-checking (AFC) as a pipeline consisting of different stages, surveyed in a variety of axes (Thorne and Vlachos, 2018; Kotonya and Toni, 2020a; Zeng et al., 2021; Nakov et al., 2021; Guo et al., 2022). However, these surveys focus on a single modality, text. This is different to real-world misinformation that often occurs via several modalities.

In AFC, the term *multimodal* has been used to refer to cases where the claim and/or evidence are expressed through different or multiple modalities (Hameleers et al., 2020; Alam et al., 2022; Biamby et al., 2022). Examples of multimodal misinformation include: $(i)$ claims about digitally manipulated content (Agarwal et al., 2019; Rössler et al., 2018) such as photos depicting former US president Trump's arrest (Figure 1); $(ii)$ combining

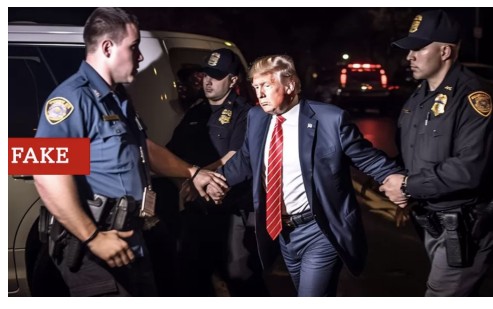

Figure 1: Manipulated image depicting arrest of former US president Donald Trump (source: BBC[1]).

content from different modalities and contexts, e.g. using video footage in a misleading context (Aneja et al., 2021; Biamby et al., 2022; Abdelnabi et al., 2022); $(iii)$ embedding a claim in another modality, e.g. a meme, an image with embedded text (Qu et al., 2022a), with notable real-world examples including a Brexit Vote Leave poster[2] and TikTok videos with COVID misinformation (Shang et al., 2021); $(iv)$ verifying a claim with evidence from a different modality than the input claim, e.g. verifying images against text (Shao et al., 2023), audio against textual metadata (Kopev et al., 2019), and text against images (Akhtar et al., 2023).

Fact-checking multimodal misinformation is important for a number of reasons. First, multimodal content is perceived as more credible compared to text containing a similar claim (Newman et al., 2012). For example, previous research shows that visual content exhibits a "photo truthiness"-effect (Newman and Zhang, 2020), biasing readers to believe a claim is true. Second, multimodal content spreads faster and has a higher engagement than text-only posts (Li and Xie, 2020). Third, with recent advances in generative machine learning models (Rombach et al., 2022), the generation of multimodal misinformation has been simplified.

To validate the importance of multimodal fact-

---

* This work was partially done during Mubashara's research visit at Cambridge.

[1]https://www.bbc.com/news/
world-us-canada-65069316

[2]https://www.itv.com/news/2019-01-18/
boris-johnson-under-attack-over-turkey-claim/

| Claim Modality | Percentage |
|---|---|
| Image | 20.07% |
| Video | 8.06% |
| Audio | 0.55% |
| Total | 28.68% |

Table 1: Non-textual modalities present and/or used in addition to text in our manually annotated snapshot of real-world claims from the Google ClaimReview API.

checking, we manually annotated 9,255 claims from the AVeriTeC dataset (Schlichtkrull et al., 2023), which were collected with the Google FactCheck ClaimReview API[3]. For each claim, we identified the modalities present in it and evidence strategies (e.g. identification of geolocation) used for fact-checking. We find that more than $2,600$ (28.68%) claims either contain multimodal data or require multimodal reasoning for verification, with 20.07% involving images, 8.06% videos, and 0.55% audios (see Table 1).[4] These claims can neither be fact-checked by a text-only model, nor by a model with no text capabilities.

In this survey, we introduce a three-stage framework for multimodal automated fact-checking: claim detection and extraction, evidence retrieval, and verdict prediction encompassing veracity, manipulation and out-of-context classification, as well as justification production. The input and output data of each stage can have different or multiple modalities. For each stage, we discuss related terms and definitions developed in different research communities. In contrast to previous surveys on multimodal fact-checking that focus on individual subtasks (Cao et al., 2020; Alam et al., 2022; Abdali, 2022), we consider all subtasks surveying benchmarks and modeling approaches for them.

We focus on four prevalent modalities of real-world fact-checking identified in our annotations: text, image, audio, and video. While tables and knowledge graphs are increasingly used as evidence for benchmarks (Chen et al., 2020; Aly et al., 2021; Akhtar et al., 2022), they have been covered in previous surveys (Thorne and Vlachos, 2018; Zeng et al., 2021; Guo et al., 2022). Finally, we discuss the extent to which current approaches to AFC work for multimodal data, and promising directions for further research (Section 4). We accompany the

survey with a repository,[5] which lists the resources mentioned in our survey.

## 2 Task Formulation

This section introduces a conceptualisation of multimodal AFC as a three-stage process, including claim detection and extraction, evidence retrieval, and production of verdicts and justifications for various types of misinformation (Figure 2). Compared to the text-only pipeline presented in Guo et al. (2022), our framework extends their first stage with a claim *extraction* stage, and generalises their third stage to cover tasks that fall under multimodal AFC, thus accounting for its particular challenges.

**Terminology.** A number of works (Singhal et al., 2022; Fung et al., 2021) use the term *multimedia*, which is also more common in public discussions instead of *multimodal* (Lauer, 2009). However in in this survey we adopt the latter, following other surveys that use multimodal data (Liang et al., 2022; Guo et al., 2019). Adopting the terminology of previous surveys (Thorne and Vlachos, 2018; Alam et al., 2022) and following advice from institutions such as the UNO (Ireton and Posetti, 2018), we avoid *multimodal fake news* (Meel and Vishwakarma, 2021; Amri et al., 2021; Patwa et al., 2022) due to the term's ambiguous use.

**Stage** 1**: Claim Detection and Extraction.** The first pipeline stage aims to find *checkable* (i.e. factually-verifiable) and *check-worthy* (i.e. important factual claims (Hassan et al., 2015b)) claims. Debunking a typical claim and writing the fact-checking article takes approximately one day for a human fact-checker (Hassan et al., 2015a). This stage aims to focus the AFC process on claims which are verifiable and most impactful. Multimodal claims can be diverse and include: (1) a written claim embedded in another modality (Prabhakar et al., 2021; Maros et al., 2021) such as an image or a spoken claim in an audio or video; (2) a claim that a piece of content is authentic, e.g. that a video footage is from a specific geographic location (Zhang et al., 2018; Heller et al., 2018); (3) a claim for which the evidence is manipulated to support it, e.g. through lip-syncing (Rössler et al., 2018). While in some cases the claim is clearly specified (e.g. in form of a headline), in often multiple modalities are required to understand and ex-

[3]https://toolbox.google.com/factcheck/apis
[4]Annotations at http://github.com/MichSchli/AVeriTeC.
[5]https://github.com/Cartus/Automated-Fact-Checking-Resources

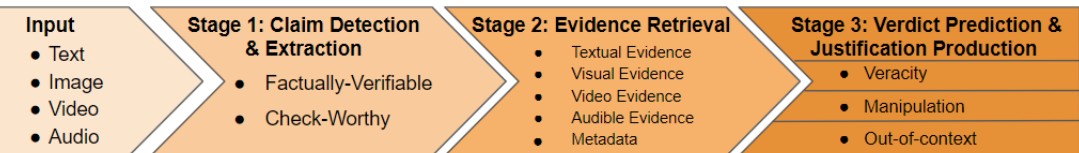

Figure 2: Multimodal fact-checking pipeline.

tract a claim at this stage. Simply *detecting* potentially misleading content is often not enough – it is necessary to *extract* the claim before fact-checking it in the subsequent stages. For example, detecting text in images or videos and understanding it given the context (Qu et al., 2022b) or verifying audios by transcribing and extracting claims (Maros et al., 2021).

**Stage** 2**: Evidence Retrieval.** Similarly to fact-checking with text, multimodal fact-checking often relies on evidence to make judgments, similar to the process followed by human fact-checkers (Silverman, 2013; Nakov et al., 2021). Two main approaches have been used in the past: $(i)$ using the claim to-be-checked as evidence itself, e.g. to detect manipulation (Qi et al., 2019; Bonettini et al., 2020); this can be seen as the multimodal version of evidence-free fact-checking of text claims by checking logical fallacies in the text (Jin et al., 2022), and $(ii)$ retrieving additional evidence (Abdelnabi et al., 2022). In multimodal fact-checking, the evidence modality can be different from the claim modality. For example, to retrieve evidence for image or audio fact-checking, previous works have also used text e.g. metadata, social media comments, or captions (Gupta et al., 2013; Huh et al., 2018; Müller-Budack et al., 2020; Kopev et al., 2019).

**Stage** 3**: Verdict Prediction and Justification Production.** Following the fact-checking process of professional fact-checkers, the final stage comprises verdict prediction and the production of justification that explains the fact-check to humans (Graves, 2018). Verdict prediction is decomposed into three tasks considering prevalent multimodal misinformation types: manipulation, using content out-of-context, and veracity classification.

**Stage** 3.1**: Manipulation Classification.** Manipulation classification commonly addresses $(i)$ misinformative claims with manipulated content; $(ii)$ correct claims accompanied by manipulated content (e.g. to increase credibility). Many methods exist to manipulate text, visual and audio content. While some require more knowledge to use (e.g. speech synthesis), other manipulations can be achieved with simple tools (e.g. changing speed of videos) (Paris and Donovan, 2019). Different terms have been used for manipulated content in recent years. A *deepfake* is commonly defined as "the product of artificial intelligence (AI) applications that [...] create fake videos that appear authentic" (Maras and Alexandrou, 2019), with popular examples including realistic-looking videos where the speaker's voice or face is modified (Paris and Donovan, 2019). On the other hand, *cheap fake* defines manipulated content created through more accessible methods (Paris and Donovan, 2019), e.g. changing captions or speed of videos (La et al., 2022). The term *fauxtography* was first coined in journalism for images manipulated to "convey a questionable (or outright false) sense of the events they seem to depict" (Cooper, 2007; Kalb and Saivetz, 2007). Other terms used in the literature for manipulated content are *fake* (Cheema et al., 2022), *forgery* (Cozzolino et al., 2021), and *splice* (Zampoglou et al., 2015).

**Stage** 3.2**: Out-of-context Classification.** Using unchanged content out-of-context is one of the most common and easiest methods to create multimodal misinformation (Luo et al., 2021; Aneja et al., 2021), and involves (possibly misinformative) textual claims paired with content (e.g. a video) taken out of context (Zhang et al., 2018; Abdelnabi et al., 2022; Garimella and Eckles, 2020). Recent work has also studied the applicability of traditional multimodal misinformation detection methods to identify out-of-context content (Zhang et al., 2023). Other terms used for combining multimodal content in a misleading way include *cross-modal (in-) consistency* (Müller-Budack et al., 2020) and *repurposing* (Luo et al., 2021).

**Stage** 3.3**: Veracity Classification.** This task is the multimodal counterpart to classifying the veracity of textual claims given retrieved evidence

(Thorne and Vlachos, 2018). Veracity classification of claims embedded in audio is also commonly referred to as *deception detection* (Kopev et al., 2019; Kamboj et al., 2021). While earlier work considered mostly claims recorded in staged setups (Newman et al., 2003) or from court trials (Pérez-Rosas et al., 2015), more recently real-world political debates have become popular. (Kopev et al., 2019; Kamboj et al., 2021).

**Stage** 3.4**: Justification Production.** Different to previous research on automated justification production (Kotonya and Toni, 2020a), human fact-checkers also give justifications for fact-checks involving images, audios, or videos (Silverman, 2013). Justifications for multimodal misinformation can be grouped in three categories: $(i)$ identifying which part of the claim input is misleading (e.g. specific areas in a visual claim or words in a textual one) (Kou et al., 2020; Purwanto et al., 2021; Lourenço and Paes, 2022); $(ii)$ providing natural language justifications following human fact-checkers (Yao et al., 2022); $(iii)$ selecting and highlighting evidence parts used for verification (Atanasova et al., 2020; Shang et al., 2022). Justifications serve purposes beyond explaining veracity classification, e.g. human fact-checkers also use them to discuss uncertainties and potential errors – especially needed in fact-checking for rapidly developing events (Silverman, 2013).

# 3 Datasets and Modeling Approaches

## 3.1 Stage 1: Claim Detection and Extraction

**Input.** Typical inputs to claim detection are uni-modal, including image (Garimella and Eckles, 2020; Qu et al., 2022a), audio (Maros et al., 2021), and video (Shang et al., 2021; Qi et al., 2022), which are collected from social media platforms such as WhatsApp and TikTok (see Table 2). The written or spoken claim is extracted from the input at this stage before fact-checking it.

**Output.** Claim detection is typically framed as a classification task. Models predict if a claim is checkable or check-worthy (Prabhakar et al., 2021; Cheema et al., 2022; Barrón-Cedeño et al., 2023). The verdict for factual-verifiability is often binary (Jin et al., 2017; Shang et al., 2021). For check-worthiness, Prabhakar et al. (2021) defines three categories of multimodal claims: statistical/numerical claims, claims about world events/places/noteworthy individuals, and other factual claims. Cheema et al. (2022) extend the binary labels for textual check-worthiness (Hassan et al., 2015b) with images to be considered as well. A tweet is considered check-worthy if it is potentially harmful, breaking news, or up-to-date.

**Modeling Approaches.** Detecting claims is a challenging task due to the vast number of posts that are published every day. Existing claim detection methods primarily rely on input content since the large volume of potentially check-worthy inputs makes it difficult to retrieve and use evidence. The early multimodal method directly concatenated visual and textual features for detection (Jin et al., 2017; Wang et al., 2018). However, simple modality fusion may not be sufficient to capture the complex relationships among multimodal information. As a result, later efforts focused on jointly learning representations across modalities. For instance, Khattar et al. (2019) leverage a variational auto-encoder (Kingma and Welling, 2014) to learn a shared representation of visual and textual content. Various attention mechanisms have also been developed to fuse multimodal representations (Qian et al., 2021; Wu et al., 2021; Liu et al., 2023b; Qi et al., 2023). Another popular approach is to use graph neural networks (Kipf and Welling, 2017) to model the interactions among different modalities (Zheng et al., 2022; Sun et al., 2023).

Multimodal content can implicitly provide claims, as seen in images and videos on social media that often have accompanying text. To extract claims from visual input, OCR systems are commonly used (Garimella and Eckles, 2020; Prabhakar et al., 2021). Qu et al. (2022b) use Google Vision API to identify text in memes. Claim extraction becomes more challenging when dealing with video inputs. Shang et al. (2021) address this challenge by extracting captions and audio chunks after sampling video frames. These captions and audio chunks were then encoded into representations to guide the visual feature extraction process. For audio inputs, Maros et al. (2021) use Google's Speech-to-Text API to produce transcripts.

## 3.2 Stage 2: Evidence Retrieval

Previous work uses different types of evidence and retrieval methods given the modalities involved. Evidence data and retrieval approaches can be grouped into $(i)$ content-based and $(ii)$ retrieval-based (see column *evidence* in Table 3).

**Content-based.** Content-based approaches use the

| Dataset | Input | Context | Output | #Input | Lang | Source |
|---------|-------|---------|--------|--------|------|--------|
| Weibo (Jin et al., 2017) | Img/Txt | Meta | 2 | 9,528 | Zh | Weibo/News |
| FauxBuster (Zhang et al., 2018) | Img/Txt | Txt/Meta | 2 | 917 | En | Twitter/Reddit |
| Exfaux (Kou et al., 2020) | Img/Txt | Txt | 2/4 | 263 | En | Twitter/Reddit |
| MuMIN (Nielsen and McConville, 2022) | Img/Txt | Meta | 3 | 12,914 | Mul | Twitter |
| MMClaims (Cheema et al., 2022) | Img/Txt | - | 4 | 3,400 | En | Twitter |
| ContrastFaux (Zong et al., 2023) | Img/Txt | - | 2 | 1,841 | En | Twitter/Reddit |
| CLEF2023 (Barrón-Cedeño et al., 2023) | Img/Txt | - | 4 | 6,000 | Mul | Twitter |
| MR2 (Hu et al., 2023) | Img/Txt | Txt/Img/Meta | 3 | 14,700 | Mul | Twitter/Weibo |
| IndiaWApp (Garimella and Eckles, 2020) | Img | Meta | 2 | 2,500 | Mul | WhatsApp |
| DisinfoMeme (Qu et al., 2022a) | Img | - | 2 | 1,170 | En | Reddit |
| WhatsApp (Maros et al., 2021) | Aud | Meta | 2 | 42,689 | Pt | WhatsApp |
| TikTok (Shang et al., 2021) | Vid | Txt/Meta | 2 | 891 | En | TikTok |
| COVID-VTS (Liu et al., 2023a) | Vid | Txt/Aud | 2 | 10,000 | En | Twitter |
| FakeSV (Qi et al., 2022) | Vid | Txt/Meta | 2 | 3,654 | Zh | TikTok/Kuai |
| MisDissem (Resende et al., 2019) | Vid/Aud/Img/Text | Meta | 2 | 121,781 | Pt | WhatsApp |
| CheckMate (Prabhakar et al., 2021) | Vid/Img/Text | Meta | 3 | 2,200 | Hi | Sharechat |

Table 2: Datasets for claim detection. Img, Txt, Vid, Aud, and Meta denote image, text, video, audio, and metadata, respectively. Output indicates the number classification labels. Mul indicates that the input has multiple languages.

claim and its context (i.e. the same information that is used for claim detection and extraction) as evidence instead of retrieving additional data. This is particularly common for audio and video misinformation (Table 3). Acoustic or visual features extracted from the input are used as evidence for verdict prediction (Wu et al., 2015; Yi et al., 2021; Ismael Al-Sanjary et al., 2016; Jiang et al., 2020). Most approaches use audio (or video) features and accompanying data (e.g. metadata, transcripts if available) as evidence to identify inconsistencies (Kopev et al., 2019; Rössler et al., 2018; Li et al., 2020b). Several datasets with image/text claims (Tan et al., 2020; Luo et al., 2021; Aneja et al., 2021) also do not retrieve additional evidence (Table 3) but rely on the given claim input or use accompanying metadata (Jaiswal et al., 2017; Sabir et al., 2018). Metadata is also often used as evidence for verdict prediction with images as input (Table 3). Jaiswal et al. (2017) and Sabir et al. (2018) use metadata (e.g. image timestamps) to provide additional information. Similarly, Huh et al. (2018) incorporate EXIF metadata (e.g. camera version, focal length, resolution settings) to detect manipulation. Image captions are also used as evidence sometimes (Shao et al., 2023).

**Retrieval-based.** Retrieved evidence external to the claim is mostly used for fact-checking text claims, text/image and image claims while audio and video fact-checks often don't retrieve additional evidence data (Table 3) but rely on the content of the video/audio input. Fung et al. (2021) leverage a knowledge base for additional background knowledge. They first construct a knowledge graph of the input news article using its text

and images. They extract entities/relations from this knowledge graph with an Information Extraction system (Li et al., 2020a; Lin et al., 2020) and map the entities to Freebase (Bollacker et al., 2008) as their background knowledge base. Two recent datasets scrape claims from fact-checking websites, and include text/image/video from those articles as evidence (Singhal et al., 2022; Yao et al., 2022). Akhtar et al. (2023) used chart images as evidence to verify textual claims. To determine if an image is used out-of-context, previous works also use *(reverse) image search* (Müller-Budack et al., 2020; Abdelnabi et al., 2022), to find evidence sources with images similar to or same as the claim image. Müller-Budack et al. (2020) query search engines and the *WikiData* knowledge graph using named entities from the claim text. Abdelnabi et al. (2022) use the claim image caption and the image itself as query.

### 3.3 Stage 3: Verdict Prediction

As introduced in Section 2, the verdict prediction stage includes manipulation, out-of-context, and veracity classification as sub-tasks.

**Input.** As shown in Table 3, inputs of **manipulation classification** datasets usually focus on one modality. For dataset creation, manipulated images are often collected from social media platforms such as Twitter, Reddit, and YouTube (Gupta et al., 2013; Heller et al., 2018). For verdict prediction datasets with videos, in addition to social media (Ismael Al-Sanjary et al., 2016), film clips (Guera and Delp, 2018), facial expressions (Rössler et al., 2018), and interviews (Li et al., 2020b) are used. Some works record videos to simulate real-world

| Dataset | Input | Evidence | Output | Tasks | #Input | Lang | Source |
|---|---|---|---|---|---|---|---|
| MAIM (Jaiswal et al., 2017) | Img/Txt | Meta | 2 | O | 239,968 | En | Flickr |
| MEIR (Sabir et al., 2018) | Img/Txt | Meta | 2 | O | 140,096 | En | Flickr |
| TNews (Müller-Budack et al., 2020) | Img/Txt | Img | 2 | O | 72,561 | En | News |
| News400 (Müller-Budack et al., 2020) | Img/Txt | Img | 2 | O | 400 | En/De | News |
| NeuralNews (Tan et al., 2020) | Img/Txt | - | 4 | O | 128,000 | En | Grover/GoodNews |
| COSMOS (Aneja et al., 2021) | Img/Txt | - | 2 | O | 201,700 | En | News/Snopes |
| NewsCLIPings (Luo et al., 2021) | Img/Txt | - | 2 | O | 988,283 | En | CLIP/VisualNews |
| InfoSurgeon (Fung et al., 2021) | Img/Txt | KB/Meta | 2 | O | 30,000 | En | VoA |
| Factify (Suryavardan et al., 2023b) | Img/Txt | Txt | 5 | O | 50,000 | En | Twitter |
| FakingSandy (Gupta et al., 2013) | Img | Txt/Meta | 2 | M | 16,117 | - | Twitter |
| MediaEval (Boididou et al., 2014) | Img | Txt/Meta | 2 | M | 13,924 | - | Twitter |
| In-the-Wild (Huh et al., 2018) | Img | Meta | 2 | M | 201 | - | Reddit/Onion |
| PS-Battles (Heller et al., 2018) | Img | Txt/Meta | 2 | M | 103,028 | - | Reddit |
| DGM (Shao et al., 2023) | Img | Txt | 2 | M | 230,000 | - | News |
| VTD (Ismael Al-Sanjary et al., 2016) | Vid | - | 2 | M | 33 | En | YouTube |
| Faceforensics (Rössler et al., 2018) | Vid | - | 2 | M | 1,004 | En | YouTube |
| DeepfakeDetect (Guera and Delp, 2018) | Vid | - | 2 | M | 600 | En | Vid Webs./HOHA |
| DFDC (Dolhansky et al., 2019) | Vid | - | 2 | M | 128,154 | En | Recorded |
| DeeperForensics-1.0 (Jiang et al., 2020) | Vid | - | 2 | M | 60,000 | En | Recorded |
| Celeb-DF (Li et al., 2020b) | Vid | - | 2 | M | 6,229 | En | YouTube |
| KoDF (Kwon et al., 2021) | Vid | - | 2 | M | 237,942 | Ko | Recorded |
| DF-Platter (Narayan et al., 2023) | Vid | - | 2 | M | 133,260 | En | YouTube |
| ASVspoof (Wu et al., 2015) | Aud | - | 2 | M | 16,375 | En | SAS |
| Phonespoof (Lavrentyeva et al., 2019) | Aud | - | 2 | M | 34,407 | En | ASVspoof |
| FoR (Reimao and Tzerpos, 2019) | Aud | - | 2 | M | 53,868 | En | TTS Systems |
| DeepSonar (Wang et al., 2020) | Aud | - | 2 | M | 18,614 | En/Zh | TTS Systems/VCC |
| HAD (Yi et al., 2021) | Aud | - | 3 | M | 88,035 | Zh | AISHELL-3 |
| FakeAVCeleb (Khalid et al., 2021) | Vid/Aud | - | 4 | M | 20,000 | En | VoxCeleb2 |
| MedVideo (Hou et al., 2019) | Vid | - | 2 | VC | 250 | En | YouTube |
| CLEF2018 Audio (Kopev et al., 2019) | Aud | Meta | 3 | VC | 286 | En | Debates |
| FactDrill (Singhal et al., 2022) | Txt | Vid/Aud/Img/Txt/Meta | 5 | VC | 22,435 | Mul | FC webs. |
| MMM (Gupta et al., 2022) | Txt | Img/Meta | 2 | VC | 10,473 | Mul | FC webs. |
| ChartFC (Akhtar et al., 2023) | Txt | Img | 2 | VC | 15,886 | En | TabFact |
| Fauxtography (Zlatkova et al., 2019) | Img/Txt | Meta | 2 | VC | 1,233 | En | Snopes/Reuters |
| MOCHEG (Yao et al., 2022) | Img/Txt | Img/Txt | 3 | VC | 21,184 | En | FC webs. |
| r/Fakeddit (Nakamura et al., 2020) | Img/Txt | Meta | 2/3/6 | O/M/VC | 1,063,106 | En | Reddit |

Table 3: Datasets for manipulation, out-of-context, and veracity classification. O, M and VC denote out-of-context, manipulation and veracity classification, respectively. Mul indicates the input has multiple languages.

scenarios (Dolhansky et al., 2019; Jiang et al., 2020; Kwon et al., 2021). To create datasets of manipulated content, altering methods based on GANs have also been applied in earlier works (Zakharov et al., 2019; Nirkin et al., 2019; Karras et al., 2019). For audio manipulations, most benchmarks (Wu et al., 2015; Kinnunen et al., 2017; Reimao and Tzerpos, 2019; Wang et al., 2020; Yi et al., 2021) use speech synthesis and voice conversion algorithms to collect manipulated audios. To assess real-world audio manipulations, Lavrentyeva et al. (2019) emulate realistic telephone channels.

Most **out-of-context classification** datasets have image-caption pairs as input (Table 3). Jaiswal et al. (2017) replace captions of Flickr images to get mismatched pairs. As replacing the entire caption can be easy to detect, later efforts (Sabir et al., 2018; Müller-Budack et al., 2020) propose to change specific entities in them. Luo et al. (2021) show that such text manipulations introduce linguistic biases and can be solved without the images. They use CLIP (Radford et al., 2021) to filter out pairs that do not require multimodal modeling. Popular sources for out of context datasets with text and image

claims include Flickr and news/fact-checking websites (Aneja et al., 2021; Jaiswal et al., 2017; Sabir et al., 2018).

The primary input to multimodal **veracity classification** is the content-to-be-checked itself – typically text, audio or video in past benchmarks. Kopev et al. (2019) include verified speeches from the CLEF-2018 Task 2 (Nakov et al., 2018) while Hou et al. (2019) collect videos about prostate cancer verified by urologists. Zlatkova et al. (2019) and Yao et al. (2022) collect viral images with texts verified by dedicated agencies. Nakamura et al. (2020) collect image-text pairs from Reddit via distant supervision, e.g. labeling a post from the subreddit "fakefacts" as *misleading* and from "photoshopbattles" as *manipulated*. For veracity classification of spoken claims, real-world political debates are popular sources for claims (Kopev et al., 2019; Kamboj et al., 2021). For example, Kopev et al. (2019) and Kamboj et al. (2021) use claims labelled by fact checking organizations, and video recordings as well as transcripts of the respective political debates.

**Output.** Most manipulation and out-of-context

classification datasets use binary labels: "out-of-context/not out-of-context" (Müller-Budack et al., 2020; Luo et al., 2021), "pristine/falsified" (Boididou et al., 2014; Heller et al., 2018), "manipulation/no manipulation" (Dolhansky et al., 2019; Li et al., 2020b). Following fact-checkers, veracity classification datasets (Singhal et al., 2022; Nakamura et al., 2020) sometimes employ multi-class labels to represent degrees of truthfulness (e.g. true, mostly-true, half-true) (see Table 3). Mishra et al. (2022) adopt labels to denote the entailment between different claim and evidence modalities, e.g. the label *support text* denotes that only the textual part of the evidence supports the claim but not the accompanying image while *support multimodal* includes both modalities.

**Modeling Approaches.** To detect visual manipulations, early approaches mostly use CNN models, such as VGG16 (Amerini et al., 2019; Dang et al., 2020), ResNet (Amerini et al., 2019; Sabir et al., 2019), and InceptionV3 (Guera and Delp, 2018). Some works extend them to capture temporal aspects of video **manipulation classification**. Amerini et al. (2019) adopt optical flow fields to capture the correlation between consequent video frames and detect dissimilarities caused by manipulation. Guera and Delp (2018) model temporal information with an LSTM model and a sequence of features vectors per video frame to classify manipulated videos. Sabir et al. (2019) similarly extract features for video frames and detect discrepancies between frames using a recurrent convolution network. Some recent models also integrate transformer-based components (Vaswani et al., 2017; Zheng et al., 2021). For example, Wang et al. (2022) combine CNNs and vision transformers (ViTs) (Dosovitskiy et al., 2021) while Wodajo and Atnafu (2021) introduce a multi-scale ViT with variable patch sizes.

Models for **out-of-context** and **veracity classification** typically consist of unimodal encoders, a fusion component to obtain joint, multimodal representations, and a classification component. To obtain text representations, early approaches used combinations of word2vec models (Mikolov et al., 2013), LSTMs (Hochreiter and Schmidhuber, 1997), and TF-IDF scores for n-grams (Jin et al., 2017; Tanwar and Sharma, 2020; Hou et al., 2019). More recent efforts use pretrained language models (Fung et al., 2021; Aneja et al., 2021; La et al., 2022). To encode visual data, many approaches

first detect objects in visual content using a Mask R-CNN model (He et al., 2017) before extracting visual features (Aneja et al., 2021; La et al., 2022; Shang et al., 2022). Visual representations for images and videos are commonly obtained using CNN models such as ResNet (He et al., 2016; Garimella and Eckles, 2020; Abdelnabi et al., 2022), VGG (Simonyan and Zisserman, 2015; Jin et al., 2017; Sabir et al., 2018), and Inception (Szegedy et al., 2015; Guera and Delp, 2018; Roy and Ekbal, 2021). To obtain audio features for voice quality, loudness, and tonality, Shang et al. (2021) extract the Mel-frequency cepstral coefficient, Kopev et al. (2019) use the INTERSPEECH 2013 ComParE feature set (Eyben et al., 2013), and Hou et al. (2019) use the openEAR toolkit (Eyben et al., 2009). Various approaches have been used to obtain **multimodal representations**. Early fusion, which joins representations immediately after the encoding step (Baltrusaitis et al., 2019) is more common (Aneja et al., 2021; Tanwar and Sharma, 2020; La et al., 2022) than late fusion (Yao et al., 2022). Moreover, model-agnostic methods (e.g. concatenation and dot product) are more prevalent (Aneja et al., 2021; Kopev et al., 2019; Jin et al., 2017; La et al., 2022) than model-based approaches (e.g. neural networks) (Jaiswal et al., 2017; Shang et al., 2022). Also popular for out-of-context classification are *cross-modality checks* that compare modalities present in a claim to each other, e.g. a video and its caption (Müller-Budack et al., 2020; Fung et al., 2021).

## 3.4 Stage 3: Justification Production

A small number of datasets is available for multimodal justification production. Previous work can be grouped into two categories: (1) highlighting parts of the input, and (2) generating natural language justifications.

**Highlighting Input.** The first category highlights input parts as justification which contribute to models' results. A popular approach for this are Graph Neural Networks (Kipf and Welling, 2017). Several papers encode multimodal data as graph elements, combining entities and their relations in and between modalities. Models are trained to detect inconsistencies between different modalities, or to detect relations (i.e., between entities) that may be misinformative. This detection could be based on the local graph structure, or on an external knowledge base (Fung et al., 2021; Shang et al., 2022;

Kou et al., 2020). Highlighted entities and relations serve as explanations for the potential misinformativeness of the entire graph. Conversely, Zhou et al. (2018) and Wu et al. (2019) use a multitask model for manipulation classification and identification of manipulated regions. Rather than labeled data, some papers rely on attention mechanisms to highlight areas as explanations. Bonettini et al. (2020); Dang et al. (2020) use this approach to highlight manipulated image regions; Purwanto et al. (2021) also include captions.

**Natural Language Justifications.** Yao et al. (2022) recently introduced a multimodal dataset with natural language justifications. They scrape text and visual content from web pages referenced by fact-checking articles. The dataset includes summaries in the fact-checking articles as gold justifications for the verdicts. However, such a setting is not realistic, as fact-checking articles are not available when verifying a new claim.

## 4 Challenges and Future Directions

**Claim extraction from multimodal content.** Multimodal claims, e.g. manipulated videos, are often embedded in specific contexts and framed as (part of) larger stories. For example, countering the misinformation in Figure 1 requires not only classifying if the image is manipulated, but understanding that it depicts the arrest of the former president in one of the cases he is being charged in. Only then can relevant evidence data be extracted and used to verify the story of Trump's arrest.To determine what is being claimed is a challenging first step in multimodal automated fact-checking. However, current efforts for multimodal claim extraction are limited to text extraction from visual content or transcribing audios and videos (Qu et al., 2022b; Garimella and Eckles, 2020; Maros et al., 2021). Addressing this challenge will require modeling approaches to effectively align and integrate all modalities present in and around the claim. For example, methods for pixel-based language modeling have recently been introduced to better align visually situated language with image content (Lee et al., 2022). Such approaches considering modalities beyond text and vision for multimodal data alignment can be useful for claim extracting from multimodal input.

**Multimodal evidence retrieval.** Evidence retrieval for audio and video fact-checking remains a major challenge. Different to other modalities,

they cannot be easily searched on the web or social media networks (Silverman, 2013). Fact-checkers often use text accompanying the videos to find evidence (Silverman, 2013). Reverse image search engines, e.g. Google Lens or TinEye, require screenshots from the video as input – and thus require the correct timeframe, which can be challenging to extract. A dedicated adversary can render current tools very difficult to use. Very often evidence for image or audio fact-checking is retrieved using text accompanying them , e.g. metadata, social media comments, or captions (Gupta et al., 2013; Huh et al., 2018; Müller-Budack et al., 2020; Kopev et al., 2019). While incorporating the textual information and the other modality (e.g. audio/image) in retrieval would provide more information, this is missing currently. How to best retrieve evidence data that is non-textual or has a different modality than the claim, also remains a challenge.

**Multilinguality and multimodality.** While there is increasing work on multilingual fact-checking (Gupta and Srikumar, 2021; Shahi and Nandini, 2020; Hammouchi and Ghogho, 2022), it is mostly limited to text-only benchmarks and models. Surveying benchmarks for different pipeline stages (Figure 2), we found limited multimodal datasets for non-English languages (see Table 3). Previous work on multilingual multimodality shows that training and testing on English data alone introduces biases, as models fail to capture concepts and images prevalent in other languages and cultures (Liu et al., 2021). Moreover, some types of multimodal misinformation exploit cross-lingual sources to mislead, e.g. images or videos from non-English newspapers appearing as out-of-context data for English multimodal misinformation (Silverman, 2013). To prevent false conclusions and biases, it is thus necessary to take approaches that are both multimodal *and* multilingual (Ruder et al., 2022). Construction of large-scale multimodal, multilingual AFC datasets would facilitate futures research in this direction, similar to benchmarks and shared tasks created for automated fact-checking tasks in English (Thorne et al., 2018; Suryavardan et al., 2023a).

**Generalizing detection of visual manipulations.** The recent popularity of diffusion models (DMs) for visual manipulation have raised questions regarding the generalizability of manipulation detectors developed for earlier models (e.g.

GANs (Goodfellow et al., 2020)). Detection models are biased towards specific manipulation models and struggle to generalize (Wu et al., 2023a; Ricker et al., 2022). A recent study (Ricker et al., 2022) shows that detectors initially developed for GANs, have average performance drops of around 15% for image by DMs. While new detection approaches for DM manipulations are already being developed (Guarnera et al., 2023; Wu et al., 2023b), the question how to generalize and increase robustness of manipulation detectors for potential future manipulation models remains open. Potential solutions can include evidence-based approaches, where the manipulated content is used to retrieve evidence data (e.g. the original video or counterfactual evidence) to prove the manipulation.

**Justifications for multimodal fact-checking.** While explainable fact-checking has received attention recently (Kotonya and Toni, 2020b; Atanasova et al., 2020), there is limited work on producing justifications for multimodal content. Previous efforts on multimodal justification production have mostly focused on highlighting parts of the input to increase *interpretability* (Kou et al., 2020; Shang et al., 2022). Natural language justifications that explain the fact-check of multimodal claims so that it is accessible to non-technical have not been developed yet. To develop solutions, we first need appropriate benchmarks to measure progress. Moreover, with the recent advances of neural models for visual and audio generation and editing, another so far unexplored direction presents itself: editing input images/videos/audios or generating entirely content to explain fact-checking results. This could include, for example, the generation of infographics or video clips to explanation fact-checks. Such a system, especially if guided by human fact-checkers (Nakov et al., 2021), would be a potent tool. As noted in Lewandowsky et al. (2020), "well-designed graphs, videos, photos, and other semantic aids can be helpful to convey corrections involving complex or statistical information clearly and concisely".

## 5   Conclusion

We survey research on multimodal automated fact-checking and introduce a framework that combines and organizes tasks introduced in various communities studying misinformation. We discuss common terms and definitions in context of our framework. We further study popular benchmarks and modeling approaches, and discuss promising directions for future research.

## Limitations

While we cite many datasets and modeling approaches for multimodal fact-checking, we describe most of them only briefly due to space constraints. Our aim was to provide an overview of multimodal fact-checking and organise previous works in a framework. Moreover, the presented survey focuses primarily on four modalities. While there are other modalities we could have included, we concentrated on those prevalent in real-world fact-checking that have not been discussed as part of a fact-checking framework in previous surveys.

## Ethics Statement

As we mention in Section 4, most datasets for multimodal fact-checking tasks are available only in English. Thus, models are evaluated based on their performance on English benchmarks only. This can lead to a distorted view about advancements on multimodal automated fact-checking as it is limited to a single language out of more than 7000 world languages. While we call for future work on a variety of languages, this survey provides an overview on the state-of-the-art of mostly-English research efforts. Finally, we want to point out that multimodal fact-checking works we cite in this survey might include misleading statements or images given as examples.

## Acknowledgements

Zhijiang Guo, Michael Schlichtkrull and Andreas Vlachos are supported by the ERC grant AVeriTeC (GA 865958). This paper is produced as part of the MuseIT project which has been co-funded by the EU under the Grant Agreement number 101061441. MuseIT has supported the work of Mubashara Akhtar. Views and opinions expressed are however those of the author(s) only and do not necessarily reflect those of the European Union or the European Research Executive Agency, REA. Neither the EU nor the granting authority can be held responsible for them.

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

## A  Methodology

We applied the following methodological approach to find and select relevant research papers for the survey.

First, after defining the research scope, we collected pivotal, highly-cited work (e.g. Nakamura et al. (2020)) and related surveys (e.g. (Alam et al., 2022)), resulting in 25 papers, as well as papers citing or cited by these works. We collected further works using the scholarly search engines Google Scholar[6], Semantic Scholar[7], DBLP[8] and ACL Anthology[9], and keyword-based search with Cartesian products of following keyword sets: {"fact checking", "fact verification", "misinformation", "disinformation", "fake news"}, {"multimodal", "text", "image", "audio", "video"}, and {"machine learning", "automated"}. The databases were queried primarily during the time frame July 26, 2022 and August 10, 2022. This step resulted in a collection of 123 papers.

---

[6] https://scholar.google.com/
[7] https://www.semanticscholar.org/
[8] https://dblp.org/
[9] https://aclanthology.org/

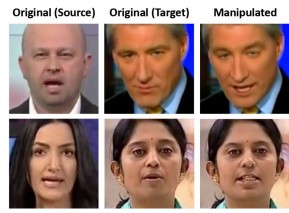

Figure 3: Example from the *FaceForensic* video manipulation dataset (Rössler et al., 2018) showing the manipulation generation approach.

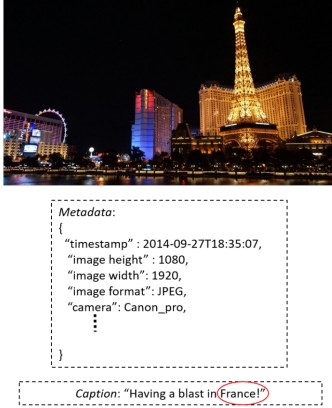

Figure 4: An entry from the *MAIM* dataset (Jaiswal et al., 2017) showing an image/text claim with metadata.

We manually screened and filtered the papers based on abstracts and introduction sections, before creating an overview of papers across the following dimensions: (1) modality; (2) fact-checking task; (3) contribution type (i.e. dataset, modeling approach, demo); (4) paper type (i.e. survey, position paper, solution paper (e.g. introducing a new benchmark or modeling approach), or evaluation paper (e.g. investigating previously proposed approaches)). Papers were mostly excluded because they focused on other tasks than fact-checking (e.g. hate speech detection) or used modalities out of our scope (e.g. tables). Moreover, during the screening process we found and added further related works, and concluded the screening with 84 unique papers.

The taxonomy of tasks (Section 2) was created in an iterative manner starting with the task labels we assigned to works during screening. As a starting point we also used taxonomies of text-only fact-checking surveys (Guo et al., 2022; Thorne and Vlachos, 2018) and adapted them for multimodal fact-checking works.

## B  Examples: multimodal misinformation

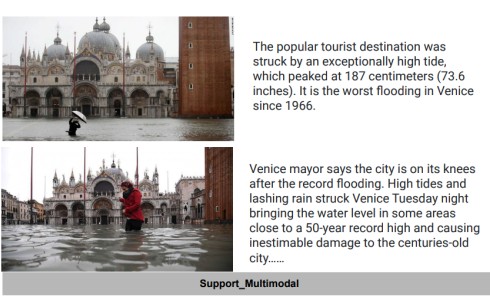

Figure 5: An entry from the *Factify* dataset (Suryavardan et al., 2023b) depicting an image/text claim and supporting image/text evidence document.

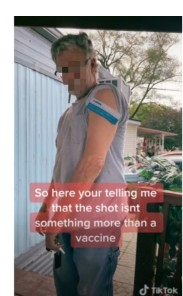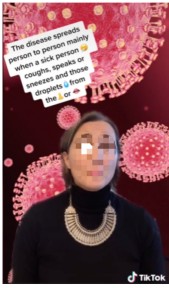

Figure 6: Left a misleading, right a non-misleading video screenshot from the Shang et al. (2021) dataset on COVID-19 TikTok Short Videos.