# OpenReview forum: "Multimodal Automated Fact-Checking: A Survey"
_EMNLP/2023/Conference — EMNLP 2023 Findings_

### Official Review · Reviewer_jMRp · 2023-07-25

**Typos Grammar Style And Presentation Improvements:** NA
**Soundness:** 3

**Excitement:**

4: Strong: This paper deepens the understanding of some phenomenon or lowers the barriers to an existing research direction.

**Missing References:**

NA

**Paper Topic And Main Contributions:**

The paper presents an extensive review of multimodal automated fact-checking tasks.
One of the most outstanding aspects of the manuscript is the vast amount of revised papers, on both, variety and quality.

Nevertheless, I think the paper lacks the structure to be clear, there are a lot of sections about specific aspects, for instance, those works related to multilinguality and multimodality, and others. This way turns into a complicated reading. Also, in line 64, the authors state "To validate the importance of multimodal fact-checking, we manually annotated 9,255 claims from.." in the rest of the paper there is no table comparing quantitative aspects of all the described and consulted works. Table 1 resume many works, but there is no comparison in terms of some experiment of quantitative comparison.

The manuscript would be more relevant if any quantitative analysis could be done, and maybe being more concise in the conclusions or define the principal insights as results of this revision.

**Questions For The Authors:**

The paper presents an extensive review of multimodal automated fact-checking tasks.
One of the most outstanding aspects of the manuscript is the vast amount of revised papers, on both, variety and quality.

Nevertheless, I think the paper lacks the structure to be clear, there are a lot of sections about specific aspects, for instance, those works related to multilinguality and multimodality, and others. This way turns into a complicated reading. Also, in line 64, the authors state "To validate the importance of multimodal fact-checking, we manually annotated 9,255 claims from.." in the rest of the paper there is no table comparing quantitative aspects of all the described and consulted works. Table 1 resume many works, but there is no comparison in terms of some experiment of quantitative comparison.

The manuscript would be more relevant if any quantitative analysis could be done, and maybe being more concise in the conclusions or define the principal insights as results of this revision.

**Reasons To Accept:**

The paper presents an extensive review of multimodal automated fact-checking tasks.
One of the most outstanding aspects of the manuscript is the vast amount of revised papers, on both, variety and quality.

**Reasons To Reject:**

Nevertheless, I think the paper lacks the structure to be clear, there are a lot of sections about specific aspects, for instance, those works related to multilinguality and multimodality, and others. This way turns into a complicated reading. Also, in line 64, the authors state "To validate the importance of multimodal fact-checking, we manually annotated 9,255 claims from.." in the rest of the paper there is no table comparing quantitative aspects of all the described and consulted works. Table 1 resume many works, but there is no comparison in terms of some experiment of quantitative comparison.

The manuscript would be more relevant if any quantitative analysis could be done, and maybe being more concise in the conclusions or define the principal insights as results of this revision.

**Reproducibility:**

4: Could mostly reproduce the results, but there may be some variation because of sample variance or minor variations in their interpretation of the protocol or method.

**Reviewer Confidence:**

5: Positive that my evaluation is correct. I read the paper very carefully and I am very familiar with related work.

---

> ### Author Rebuttal · Authors · 2023-08-29
>
> We thank the reviewer for their feedback and questions raised which we aim to address below in much detail.
>
> **Structure and sections about different aspects**
>
> We tried to organize the survey in a structured way by doing the following:
>
>     * Providing a figure with all stages and the (sub-)tasks per stage
>     * Splitting task formulation according to the stage in Fig 2. and numbering subtasks accordingly e.g. 3.1, 3.2, etc, for tasks of stage 3.
>     * splitting dataset and model discussion in Stages as well
>     * splitting dataset discussion in different paragraphs (i.e. input vs. output)
>     * providing a table on benchmarks and grouping them according to some attributes (e.g. tasks, modalities, source, etc.)
>
> If the reviewer could provide  specific suggestions on how to improve the structure, we are happy to adjust the paper accordingly.
>
> **Details and discussion about the annotated data**
>
> Due to space limits we excluded the annotated data table and discussion from the paper but definitely agree with the review that is important information.
>
> Below you can find a table highlighting the modalities of the annotated data, we will add it to the main paper with a discussion:
>
> | Claim Modality | Percentage |
> |----------------|------------|
> | Image          | 20.07%     |
> | Video          | 8.06%      |
> | Audio          | 0.55%      |
> | Total          | 28.68%     |
>
> **Quantitative comparison/experiments**
>
> We thank the reviewer for bringing these points to our attention. Our method, which involves comparing various models through the outcomes documented in their papers and discussing their similarities/ contrasts, is in line with prior surveys of the field. To illustrate, both [1] and [2], which are commonly-cited surveys in the domain of automated fact-checking, employ a similar strategy.
>
> We agree that comparative evaluations are useful, and indeed these occur in the context of shared tasks that conduct them. Many of them have proceedings such as FEVER and FACTIFY which have overview papers for this purpose. We will cite those that are related to multi-modality if accepted.
>
> **Conclusion and insights**
>
> In the Challenges and Future Directions section (L568-L687), we discuss challenges related to claim extraction from multimodal content, generalizing detection of visual manipulations, multimodal evidence retrieval, challenges related joint multimodal and multilingual fact checking, justifications for multimodal fact-checking. Moreover, we highlight insights we gained which can lead to the following future works:
>
>     * effectively align and integrate all modalities present in and around the claim for claim understanding (L586 ff.)
>     * retrieving non-textual evidence (L614)
>     * requirements for multimodal fact-checking benchmarks to facilitate future work (L622 ff.)
>     * etc.
>
> **References**
>
> [1] Thorne and Vlachos(2018): Automated fact checking: Task formulations, methods and future directions.

---

### Official Review · Reviewer_4pKp · 2023-08-05

**Soundness:** 2

**Excitement:**

4: Strong: This paper deepens the understanding of some phenomenon or lowers the barriers to an existing research direction.

**Paper Topic And Main Contributions:**

This paper conceptualises a framework for multimodal automated fact-checking. The authors talk about four modalities (e.g., text, image, audio, and video) that misinformation can be spread in the wild and presents existing benchmarks and models.

**Reasons To Accept:**

This paper well presents the motivation for multimodal (including text) automated fact-checking research and clearly organizes the paper. The citations and references are informative and comprehensive enough to represent the sub-tasks.

**Reasons To Reject:**

The authors point out a conceptualization of multimodal AFC as their contribution. As far as I am concerned, the three-stage framework of claim detection, evidence retrieval, and verdict prediction is not new to the fact checking field (https://arxiv.org/abs/1803.05355) and has been studied. Is there a justification for multimodal fact-checking should be done in this fashion? Please provide reasons why the authors consider this as the most ideal approach.

For extracting claims to be verified, this may not be the case for some data in the wild. For example, for fact-checking text associated with video content (e.g., youtube videos and their headlines), the headlines themselves are the verifiable claims that need not be verified. Is the first stage absolutely necessary for all multimodal tasks?

It is tricky to collect datasets from the wild or create artifacts that represent the natural phenomena of multimodal fact-checking due to inherent subjectivity and expensive crowdsourcing. Thus, mentioning how the benchmarks are created might be worthwhile to discuss to explain the characteristics of each dataset.

It might be worthwhile to include LLM-based detection models in the model section to reflect the current trend of fact verification and multimodal fact checking.

**Reproducibility:**

4: Could mostly reproduce the results, but there may be some variation because of sample variance or minor variations in their interpretation of the protocol or method.

**Reviewer Confidence:**

3: Pretty sure, but there's a chance I missed something. Although I have a good feel for this area in general, I did not carefully check the paper's details, e.g., the math, experimental design, or novelty.

---

> ### Author Rebuttal · Authors · 2023-08-29
>
> We thank the review for their detailed review, feedback, and questions which we address below in more detail.
>
> **Conceptualization of the multimodal AFC (the framework): relation to previous work e.g. Thorne et al. (2019) and why this is the ideal approach.**
>
> Our multimodal automated fact-checking framework is an extension of the three-step fact-checking pipeline introduced in fact-checking surveys for textual data (Thorne and Vlachos, 2018; Guo et al., 2022).
>
> While we include additional tasks specific to multimodal fact-checking (e.g. claim extraction, manipulation, out-of-context detection), we tried to align multimodal fact-checking to standard fact-checking. This allows readers familiar to standard fact-checking literature understand fact-checking in context of multiple modalities. Moreover, this allowed us to find synergies (e.g. tasks in multimodal fact-checking which overlap to some extent with standard fact-checking, such as evidence retrieval) and refer to previous surveys and literature.
> Finally, we discussed stages of the pipeline which are different from standard fact-checking pipelines, e.g. the first and final stage, in more detail and highlighted differences specific to multiple modalities (more details on that in Sec 4).
>
> **Necessity of the first stage for all multimodal tasks.**
>
> As we briefly mention in L146, the extraction of claims is often necessary from a context but we definitely agree with the review that this is not always the case (for example in the headline case mentioned by the reviewer). We will state this more clearly in the discussion of “Stage 1” in Section 2. Thank you for pointing this out!
>
> **Details on benchmark creation.**
>
> While we include various pieces of  information on the benchmarks in Table 1, such as the data source, the language, etc. - more details on how the benchmarks were created can be very insightful.
>
> We have started extracting the following benchmark creation attributes for the 36 benchmarks listed in Table 1: rationale for dataset creation, preprocessing steps conducted on data collected from “source” (last column of Table 1 currently), annotation process and demographic of annotators, limitations and potential artifacts/biases of the benchmark mentioned by their authors. A subset of the new columns is provided below, we will update accordingly in the revised paper.
>
> | Dataset Name | Rationale | Preprocessing Steps | Annotation Process and Demographic of Annotators | Limitations & Biases |
> |--------------|-----------|---------------------|------------------------|----------------------|
> | NewsCLIPpings| Explore techniques for creating challenging image-caption matches. | Derived from VisualNews. Used spaCy for named entities, REL for entity linking, SBERT-WK & CLIP for text embeddings, Faster R-CNN & CLIP for image embeddings. | Unknown | Reliance on specific pre-trained models like CLIP; Pre-determined thresholds for similarity. |
> | Factify      | Determine similarity between news tweets to classify them based on text and image similarity. | Data collected from Twitter handles; Used Sentence BERT for text similarity, Histogram and ResNet50 for image similarity. Text and images preprocessed and converted to embeddings and similarity metrics. | Unknown | Dependence on pre-decided thresholds for similarity; Reliance on specific pre-trained models. |
> | Fauxtography | Address the truthfulness of claims related to images. | Images and claims collected from Snopes' Fauxtography and Reuters' Pictures of the Year. | Images labeled based on veracity. | Imbalance in the class distribution; Variation in claim text length. |
>
> We concentrate on these benchmark attributes inspired by previous work on dataset documentation, e.g. Datasheets for Datasets (Gebru et al., 2018) and Data Cards by Huggingface.
>
> We will add a separate subsection to Section “Datasets and Modeling Approaches” discussing this information.
>
> **Large LM-based verification for multimodal data**
>
> We thank the reviewer for pointing this out to us. We would appreciate it if the reviewer can point towards some particular paper they would like us to add.
>
> While we discuss in L495-L525 different approaches for representation extraction and multimodal fusion and include LMs for text data L496, we will add more references on the recent work on LLM-based visual feature extraction (Yao et al. (2023)) and LLM-based modality fusion (e.g. Zhou et al. (2022) and Jiang et al., (2023)) for fact-checking those mentioned already.
>
> **References:**
>
> Jiang et al. (2023) Similarity-Aware Multimodal Prompt Learning for Fake News Detection
>
> Zhou et al. (2022): Multimodal Fake News Detection via CLIP-Guided Learning
>
> Thorne and Vlachos(2018): Automated fact checking: Task formulations, methods and future directions.
>
> Guo et al. (2022): A Survey on Automated Fact-Checking
>
> Gebru et al. (2018): Datasheets for Datasets
>
> Yao, Barry Menglong, et al. "End-to-end multimodal fact-checking and explanation generation: A challenging dataset and models." Proceedings of the 46th International ACM SIGIR Conference on Research and Development in Information Retrieval. 2023.

---

### Official Review · Reviewer_p4Yf · 2023-08-06

**Soundness:** 2

**Excitement:**

2: Mediocre: This paper makes marginal contributions (vs non-contemporaneous work), so I would rather not see it in the conference.

**Paper Topic And Main Contributions:**

The paper presents a detailed survey on the emerging field of multimodal automated fact-checking, filling a critical gap in the existing research landscape.

**Questions For The Authors:**

1. What are the difficulties and challenges of multimodal fact checking compared to traditional fact checking

2. The authors should further clarify which benchmarks support which stages of checking.



**Reasons To Accept:**

The paper conducts a comparative analysis of different methods and approaches for multimodal fact checking.

**Reasons To Reject:**

1. There is no any specific comparative analysis of the surveyed models or their performance on the benchmark datasets. The absence of such analysis weakens the survey's ability to provide insights into the relative effectiveness of different approaches.

2. A more in-depth discussion on the challenges and practical implications of deploying multimodal automated fact-checking in real-world settings would enhance the paper's significance.

3. Given the complexity and rapid development in the field of multimodal fact-checking, the paper appears to lack sufficient coverage of various multimodal approaches.


**Reproducibility:**

N/A: Doesn't apply, since the paper does not include empirical results.

**Reviewer Confidence:**

3: Pretty sure, but there's a chance I missed something. Although I have a good feel for this area in general, I did not carefully check the paper's details, e.g., the math, experimental design, or novelty.

---

> ### Author Rebuttal · Authors · 2023-08-29
>
> We thank the reviewer very much for their review and questions, which we aim to address in detail below.
>
> **Regarding comparative analysis of the surveyed models**
>
> We thank the reviewer for bringing these points to our attention. Our method, which involves comparing various models through the outcomes documented in their papers and discussing their similarities/ contrasts, is in line with prior surveys of the field. To illustrate, both [1] and [2], which are commonly-cited surveys in the domain of automated fact-checking, employ a similar strategy.
> We agree that comparative evaluations are useful, and indeed these occur in the context of shared tasks that conduct them. Many of them have proceedings such as FEVER and FACTIFY which have overview papers for this purpose. We will cite those that are related to multi-modality if accepted.
>
> **Regarding practical implication of deploying multimodal automated fact-checking models in real-world settings**
>
> Studying applicability of automated fact-checking in real-world settings is important for understanding the usability of models developed in research settings.
>
> For example, a recent paper by [2] (which we refer to at different places in our survey e.g. L681) focuses specifically on this aspect. The authors study available automated fact-checking tools that can support the human expert in the different steps of her fact-checking. One of the insights gained is the need to further develop multimodal datasets and technologies for automated fact-checking.
>
> This aspect is also studied and discussed in previous work on automated fact-checking from the journalism/fact-checking community. For example, in [3], misinformation verification in the real-world is discussed through multiple articles written by professional fact-checkers and journalists. The articles discuss real-world implications of fact-checking and give practical advice and guidelines to fact-checkers. For example, an article by Trushar Barot, the paper highlights the role of user generated content in news reporting, as demonstrated by BBC News during the July 2005 bombings in London. The article underscores the importance of verifying images received from the public, detailing a comprehensive process that includes establishing the image's originator, corroborating the location, date, and time of the image, ensuring its contextual accuracy, and obtaining necessary permissions.
>
> We will include a more in-depth discussion on this point (while also referring to previous work e.g. from [2] and the fact-checking community) in Section 4.
>
> **Regarding coverage and rapid development of the field**
>
> We agree with the reviewer that multimodal automated fact-checking is a rapidly evolving field. We believe our list of paper is up-to-date considering the time of submission, but we will search the literature for more recent papers, and we welcome suggestions.
>
> **Question:** *What are the difficulties and challenges of multimodal fact checking compared to traditional fact checking?*
>
> There are multiple challenges specific to multimodal fact-checking. We discuss this in Section 4 in more detail. Some of them are the following:
>
> Multimodal claims, such as manipulated videos, are deeply contextual and often form part of more extensive narratives. Effective fact-checking involves not only identifying manipulations but also understanding the communicated claim in a broader context. Current methods for extracting such claims from multimodal content while incorporating the context, focus strongly on text and audio-video transcription. (Sec 4.1)
>
> Evidence retrieval for modalities other than text and image (e.g. audio and video) is a significant challenge, as these modalities aren't easily searchable in various platforms and the web. Currently, fact-checkers frequently rely on accompanying text, such as metadata, comments, or captions, to gather evidence. While tools like reverse image search engines exist, they require precise inputs, like specific video screenshots, which can be hard to determine. (Sec 4.2)
>
> Despite the growing interest in multilingual fact-checking, most efforts remain constrained to text-only benchmarks and models. A survey of benchmarks showed a small number of multimodal datasets for non-English languages. Relying solely on English data when training and testing introduces biases, as it might not encompass concepts unique to other languages and cultures. (Sec 4.3)
>
> For further details, we refer to Section 4 of the survey.
>
> **Question:** *The authors should further clarify which benchmarks support which stages of checking.*
>
> We include this information in Table 1 under column “Tasks”, but to make this point clearer for the reader we can highlight it for each paper we mention in the main text. Thanks for the suggestion!
>
> **References**
>
> [1] James Thorne and Andreas Vlachos. 2018. Automated Fact Checking: Task Formulations, Methods and Future Directions. In Proceedings of the 27th International Conference on Computational Linguistics, pages 3346–3359, Santa Fe, New Mexico, USA. Association for Computational Linguistics.
>
> [2] Nakov, Preslav, et al. "Automated fact-checking for assisting human fact-checkers." arXiv preprint arXiv:2103.07769 (2021).
>
> [3] Craig Silverman. (2014). Verification Handbook: An Ultimate Guideline on Digital Age Sourcing for Emergency Coverage, European Journalism Centre.

---

### Meta-Review · Area_Chair_br6o · 2023-09-19

**Recommendation:** 3

**Metareview:**

The paper presents a survey of recent work in multi-modal fact checking and a framework for categorizing the sub-tasks and existing datasets. All reviewers agreed that a survey of this kind would be useful for the community. The paper covers an extensive set of previous work and the reviewers mostly agree that the coverage is mostly comprehensive.

Several of the objections come from disagreements about the importance of evaluating the performance of existing models on the benchmark datasets. However, the authors argue in their rebuttal that this is in line with previous surveys that have been published in the field.

A second objection raised by the reviewers is in the structure and organization of the paper, and more specifically, the links between specific benchmark datasets and the sub-tasks they are useful for. However, these objections are not well fleshed out in the reviewer comments. The authors have proposed a reorganization of the paper to improve its structure, as well as an extensive update to Table 1 to include more information about the rationale and process behind each dataset.

---

### Decision · Program_Chairs · 2023-10-07

**Decision:**

Accept-Findings

**Comment:**

The paper presents a survey of recent work in multi-modal fact checking and a framework for categorizing the sub-tasks and existing datasets. All reviewers agreed that a survey of this kind would be useful for the community. The paper covers an extensive set of previous work and the reviewers mostly agree that the coverage is mostly comprehensive.

Several of the objections come from disagreements about the importance of evaluating the performance of existing models on the benchmark datasets. However, the authors argue in their rebuttal that this is in line with previous surveys that have been published in the field.

A second objection raised by the reviewers is in the structure and organization of the paper, and more specifically, the links between specific benchmark datasets and the sub-tasks they are useful for. However, these objections are not well fleshed out in the reviewer comments. The authors have proposed a reorganization of the paper to improve its structure, as well as an extensive update to Table 1 to include more information about the rationale and process behind each dataset.